# A Summary of Hail Events during the Summer of 2022 in Catalonia: A Comparison with the Period of 2013–2021

**Tomeu Rigo** †🆔 **and Carme Farnell** *,†🆔

Servei Meteorològic de Catalunya, C/Berlin, 38-46, 08029 Barcelona, Catalunya, Spain
* Correspondence: carme.farnell@gencat.cat
† These authors contributed equally to this work.

**Abstract:** Hail events are common in Catalonia during the warm season (May to September), but especially between June and August. These cases produce important damages to agriculture and infrastructure. The campaign of 2022 will be remembered by three different phases: the first and last phases, which were very stable and with few events, and the middle phase, which had a large number of episodes. Some of the cases had an important social impact because of the large areas affected or the economical damages. The present analysis used the vertically integrated liquid radar product for estimating the hail swaths. Hail swaths are classified according to different parameters, allowing for the characterization of the campaign and a comparison with the period of 2013–2021. The results show how the month of June had a deficit of cases with respect to the reference period (half of the cases), July presented similar values, and August had a positive anomaly, with five times more cases. In addition, the first ever case of giant hail in Catalonia occurred in August 2022, a month with more than five times the number of cases of severe and very large hail with respect to the average of the period of 2013–2021.

**Keywords:** hail swaths; Catalonia; 2022; comparison; vertically integrated liquid (VIL); hail reports

## 1. Introduction

The summer of 2022 will be remembered in Catalonia by three meteorological elements [1]: first, it was dry or very dry in the larger part of the territory; second, it was hot or very hot in all of Catalonia (the hottest summer in history since measures have been available); and, third, the large number of severe weather events, mainly hail. These events produced important damages to infrastructure (mainly roofs, with damages of over EUR 5.6M on 30 August alone), agriculture (in areas of the interior and also, noticeably, in the coast), and forests (several pine areas were affected by Diplodia pinea), but also with some injuries and one casualty.

Catalonia has been historically affected by summer hailstorms. Because of this, many studies has analyzed the events from different perspectives: study cases [2–7]; using weather radar for diagnosing hail in thunderstorms [8–14]; thorough thermodynamics [15,16]; by means of lightning data and the lightning jump as a forecaster [17–20]; considering hail-pad data [21,22]; or determining the synoptic patterns [23].

All of the previously cited studies have allowed for an improvement in the knowledge of the behavior of hail storms in Catalonia from the point of view of forecasting, diagnosis, and real-time surveillance. However, some of the events of the summer of 2022 have been extraordinary for different causes: the region of occurrence, the registration of giant hail (10 cm diameter or more), the difficulty in the forecasting, or the total area affected, which are issues that must motivate new research for investigating whether we are at the beginning of a new nature of hail-bearing thunderstorms. In fact, other regions of the Mediterranean have recently suffered giant hail events for the first time [24,25], some European countries have been hit by recurrent large hail events [26,27], and an increase in these types of events has been reported in the United States in recent years [28,29].

The present analysis has, as a novelty, a description of the summer 2022 hail campaign in order to determine whether it was extraordinary compared with the 2013–2021 period. Then, in order to reach this goal, the first objective was the characterization of the campaign thorough radar fields and hail reports. This helped to characterize the events and to provide information about the spatial and daily, monthly, and yearly behaviors. The second objective was a comparison of the previous results with the ones for the 2013–2021 period in order to determine how extraordinary the year was and, furthermore, if all three months (June, July, and August) presented similar conditions. The manuscript continues with the presentation of the data, methodology, area of study, and, later, the main results. Finally, we discuss the main results, which are summarized in the conclusions.

## 2. Data and Methods

### 2.1. Area and Period of Study

Figure 1 shows the location of Catalonia, the area of study, included in the red rectangle. It is a region with a complex topography that borders the Mediterranean Sea, which generates milder atmospheric conditions, as has been presented in [30]. The period of study is the summer months (June, July, and August) of 2022. These months were compared with the same seasonal period for the years 2013 to 2021. We selected this yearly period because the radar database has had the same format since 2013. The monthly selection can be found in the bibliography's events from this season (the most well-known is that studied by Farnell et al. [6] or Pineda et al. [5]). However, the influence of these cases in the yearly statistics are negligible according to the previous analysis of Farnell and Rigo [17] or Rigo and Farnell [19].

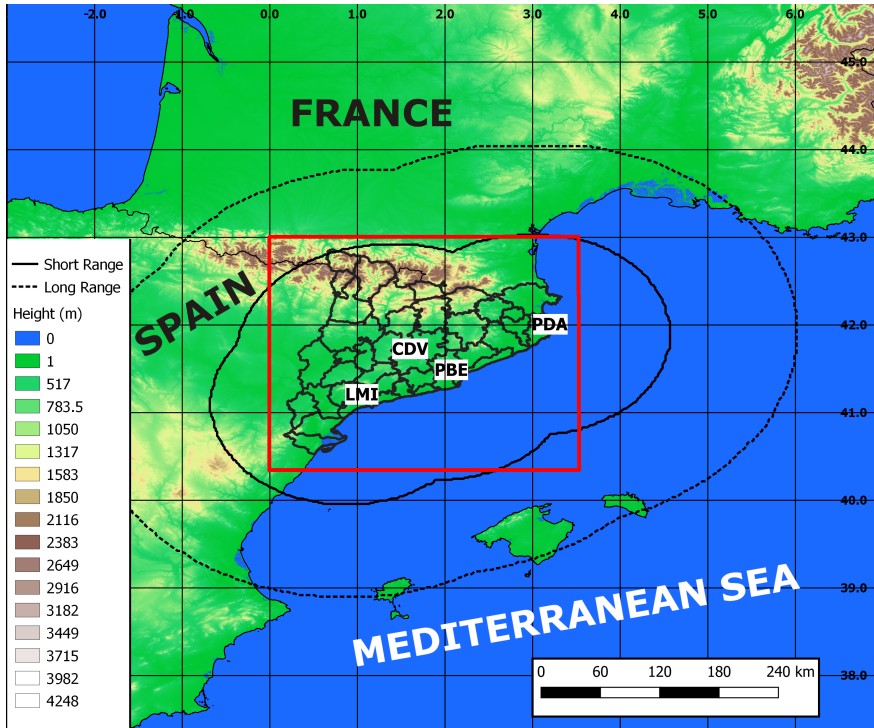

**Figure 1.** Topographic map of the area of study (marked with the red rectangle). The three-letter acronym labels (CDV = Creu del Vent, PBE = Puig Bernat, PDA = Puig d'Arques, and LMI = La Miranda) indicate the location of the radars. The short (straight) and long (dashed) radar ranges are indicated by the black lines.

### 2.2. Data Used

#### 2.2.1. Hail Reports

The hail reports correspond to the Severe Weather database of the Servei Meteorològic de Catalunya (hereafter, SWDB). The SWDB contains severe phenomena (hail over 2 cm, straight winds of 25 m/s or more, and tornadoes) reported in Catalonia since 2004. There are also other adverse (but non-severe) events (such as wide-affected-area hail cases or damaging convective winds). Each report contains the date and time, the location, and the magnitude (hail size, wind speed, or Fujita scale level) among the source of information. Farnell and Rigo (2020) [20] provides a detailed description of the SWDB. In this research, only hail reports were considered.

#### 2.2.2. Radar Fields

Rigo and Farnell (2019) [10] showed the ability of the vertically integrated liquid (VIL) product to delimit the area hit by hailstorms. Since 2013, the Servei Meteorològic de Catalunya (hereafter, SMC) has archived this product in Geotiff format for the composition of the four radars of the network (see Figure 1). The time and spatial resolutions are 6 min and $1 \times 1$ km$^2$, respectively. Because the VIL is a volumetric product, i.e., it needs data at different levels over a point, the field coverage coincides with the solid black line (short-range coverage) of the radars. This coverage ranges from 3 to 130 km from each radar. Finally, it is worth noting that the radars are C-band single-polarization systems. This fact has limitations in hail events, with the beam signal attenuation, but is minimized by the location of the radars, which means that all hailstorms in Catalonia are detected by at least two radars (see [31] for more details).

When computing integrated liquid data (VIL), the output shows the estimated precipitation (in kg/m$^2$) contained within the user-defined layer, which, in this research, corresponds to the whole radar column. The VIL algorithm first searches out those points in the layer over a given range and azimuth that intercept the plan position indicator (PPI) scans, including one point above and below. Next, the algorithm converts the reflectivity values to water content, and integrates the values in the layer. Each data point is assigned a weighting corresponding to the height interval that it represents in the layer. The result is an intermediate PPI product that has the total water content as a function of the surface range and azimuth. Finally, the intermediate product is transformed into Cartesian and stored [32,33].

### 2.3. Methodology

#### 2.3.1. Characterization of Hail Reports

Hail reports were used for identifying differences in their behavior in the summer of 2022 to those from the period 2013–2021. In this way, the registers were firstly grouped by day. Each hail day was characterized by the date, the number of observations, and the maximum diameter. The following step was to merge the daily records by months and years, including in both classifications the number of observations per period, the maximum hail recorded, and the number of days with hail, with severe hail ($\geq$2 cm), and with very large hail ($\geq$4 cm). To perform the comparison between 2013–2021 and 2022, we estimated the box plot for each variable. This comparison allows us to determine whether 2022 was an extraordinary year in terms of the number of cases, the maximum hail recorded, or the number of cases with very large hail. Any significant change was in the hail data collection during 2022 with respect to the previous campaigns. In addition, quantitative statistical testing was also performed between the data of both periods, involving a *t*-test.

2.3.2. Hail-Swaths Based on Radar Data

To determine the spatial characterization of both the 2013–2021 and 2022 periods, we used the daily maximum VIL fields and three different thresholds: 20, 40, and 60 kg/m$^2$. These thresholds allowed us to identify the areas hit by hail. A hail swath is defined as the region included behind the 20, 40, and 60 kg/m$^2$ isolines of VIL. The differences between the thresholds provide more confidence to the largest ones, which are more restrictive and, at the same time, are usually associated with the biggest hail. However, as Edwards and Thompson [34] or Blair et al. [35] showed, this product can provide similar values for a wide spectrum of hail sizes. In addition, there exists a high dependence on the season, and the vertical development of the thunderstorm and the distance of it from the radar. These limitations were observed by Rigo and Farnell [10] in the preliminary analysis, which compared this parameter with the affectation to agriculture.

Each area or hail swath is characterized by the date, the area, the coordinates of the center, and the mean and maximum VIL. In the current study, the hail swaths were used for determining the spatial degree of affectation along the Catalan territory. This allows us to know if the summer of 2022 had a larger extent than usual (which is understood as the period of 2013–2021). Finally, the spatial resolution of the fields is different according to the VIL threshold: $0.04° \times 0.04°$ for 20 mm, $0.08° \times 0.08°$ for 40 mm, and $0.12° \times 0.12°$ for 60 mm. These changes allow us to uniform the spatial differences in the lower extent of the hail swaths as the threshold increases.

## 3. Results

### 3.1. Differences in the Yearly Distribution of the Hail Reports

Figure 2 shows the different parameters from a yearly perspective in order to compare the general behavior for the summer events that occurred during the period 2013–2021 and the 2022 ones. Panel A presents the number of observations. The red box indicates that 2022 had more registers (125) than the usual number, which is 90 (the black thick line corresponds to the percentile 50 -P50-, and the top and bottom limits of the box to the percentiles 75 -P75- and 25 -P25-, respectively). However, the number is far from the maximum, 210, represented by the fine horizontal line over the box. This large value for the summer of 2022 can be initially explained by the fact that there were some cases that affected populated regions, permitting the registration of more observations than other cases that occurred in mountainous or less inhabited areas. In addition, the *t*-test for the two sets (period of 2013–2021 and 2022, respectively) gave a t value of −2.0029, a degree freedom value of 26.162, and a *p*-value of 0.05565, which is slightly higher than 0.05, meaning that it is slightly out of the limits of significance.

Moving to the maximum hail size, panel B shows one of the exceptional statistics of the summer of 2022. The value of 10 cm exceeds the maximum for the comparative period by 4 cm. It is important to indicate that we consider only those confirmed values (with a photography of the stone and some object of reference), with larger values being possible in many cases. In fact, the maximum absolute value of the SWDB until 2022 was 7 cm (recorded in 2012). In addition, through newspapers, we noticed a case of 10 cm hail stones in 2002, but there is no official confirmation.

Panel C provides more information than that indicated by panel A, because it explains if the number of registers corresponds to few or many cases. It can be observed that, for the three categories (hail, severe hail, and very large hail days), the values for 2022 were exceptional, particularly for the severe (15) and very large hail (10) days. In these categories, 2022 values constitute new records in Catalonia (with the same value as that in 2020 for the case of severe hail). On the contrary, the number of hail days (26) is slightly far from the maximum (34, in 2018).

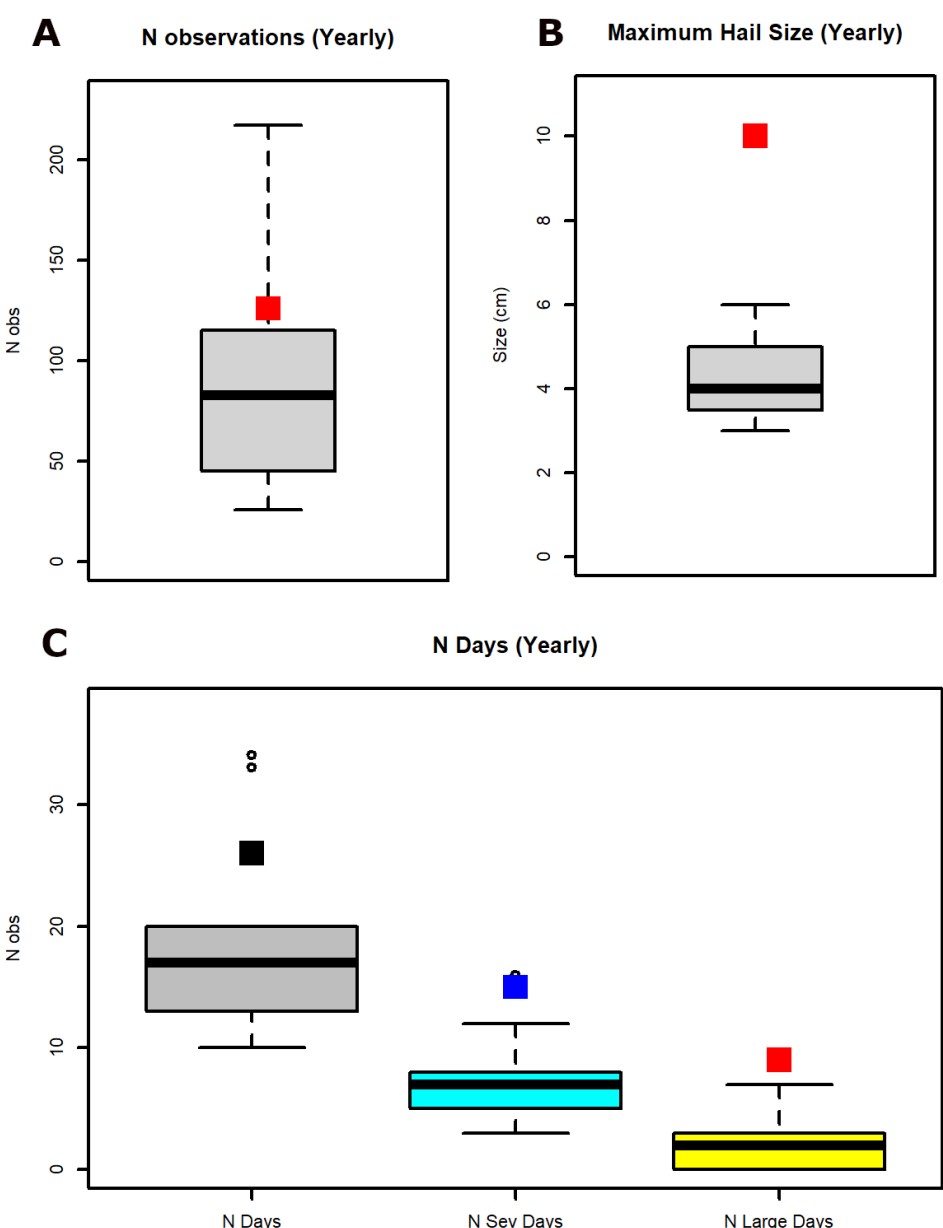

**Figure 2.** Box plots of the yearly values for the period 2013–2021 of (**A**) the number of observations, (**B**) the maximum hail size recorded, and (**C**) the number of days (from **left** to **right**: total, with severe hail, and with very large hail). The squares (red in panels **A**,**B**, and black, blue and red in panel **C**) indicate the values for the year 2022.

*3.2. Differences in the Monthly Distribution of the Hail Reports*

The monthly differences can help us to understand if the absolute values of 2022 were the product of an isolated case (which is possible in the case of phenomena such as hail, which is a stochastic random process; [36,37]) or, on the contrary, were a consequence of a generalized atmospheric situation. The general behavior of the period 2013–2021 (panel A of Figure 3 showed the maximum number of registers during the months of June (JUN) and July (JUL), whereas there was a notable decrease in August (AUG). This situation is the opposite of 2022: the number of cases in June was very scarce, July was at the P50, and there was a new record in August, with six events more than the previous maximum (2020).

A similar pattern is observed in the case of the maximum hail size for 2022 (panel B): a clear increase in the values from June (1.5 cm) to August (10 cm). However, this agrees

with the general trend for the period 2013–2021: hail stones are usually larger in August, probably because it has the warmest atmospheric conditions; [38,39].

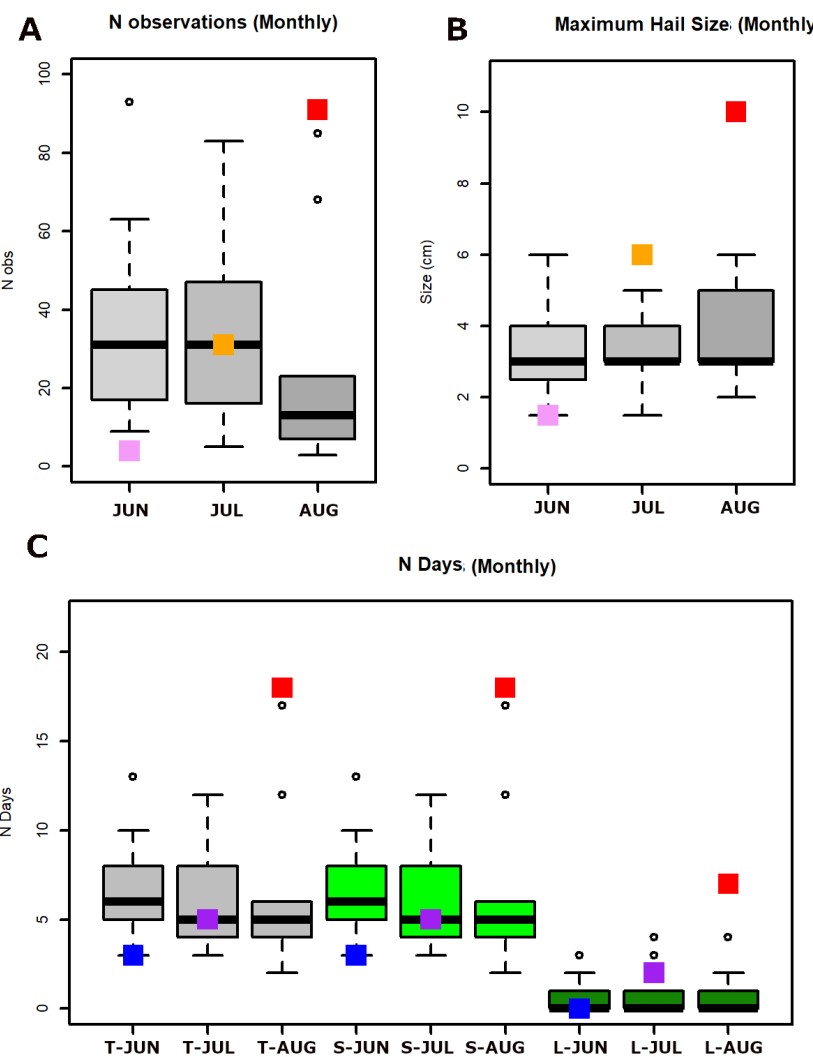

**Figure 3.** Box plots of the monthly values (only summer), for the period of 2013–2021, of (**A**) the number of observations, (**B**) the maximum hail size recorded, and (**C**) the number of days (total in grey, with severe hail in green, and with very large hail in dark green. From **left** to **right**: June, July, and August). In panels **A,B**, the light grey boxes correspond to June, grey to July, and dark grey to August. The squares (pink for June, orange for July, and red for August in panels **A,B**, and blue for June, purple for July, and red for August in panel **C**, repeated from **left** to **right** for total, severe, and very large hail days) indicate the values for the year 2022.

Finally, the number of days (panel C) again shows the exceptional August of 2022, with more hail days and severe hail days (both with 18 of the 31 days of the month) and very large hail days (7). July was in the average (except for the very large hail days), whereas June was clearly below even the P25 for the three categories. It is important to comment that the very large hail days in July occurred at the end of the month (the days 28 and 29), which is indicative that they were the start of the exceptional period.

### 3.3. Differences in the Yearly Spatial Distribution

To make a better comparison between the two periods (which have notable differences in the values because of the variation in the length), we divided the pixel values into three categories: few cases (corresponding to a unique observation), a medium number

(two for the period of 2022 and two to three for 2013–2021), and a high number (more than two for 2022 and more than three for 2013–2021). The yearly distribution (Figure 4) reveals important differences in the spatial patterns, which are more evident when the VIL threshold increases.

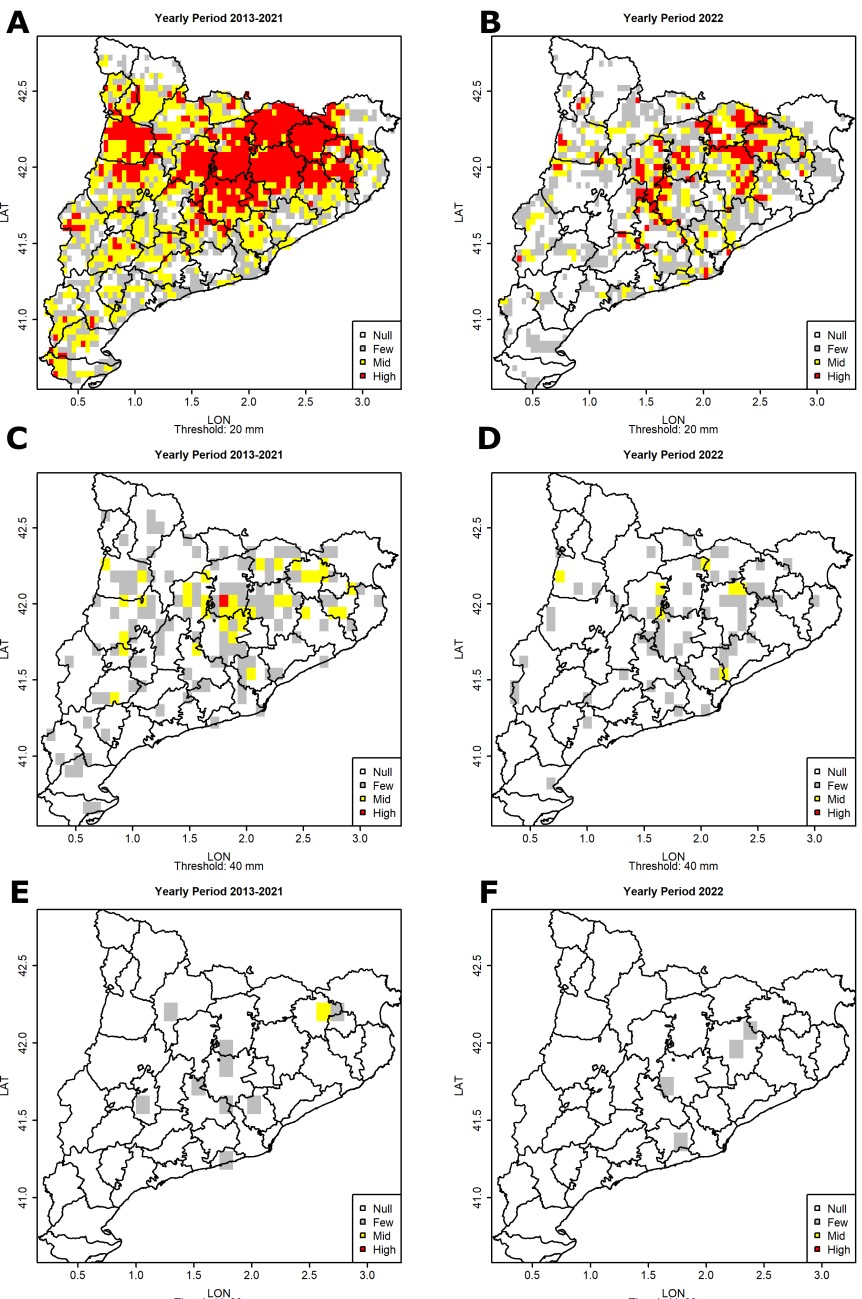

**Figure 4.** Spatial yearly distribution of the hail swaths for the period of 2013–2021 (panels **A,C,E**) and 2022 (panels **B,D,F**), for the 20 kg/m$^2$ (**A,B**), 40 kg/m$^2$ (**C,D**), and 60 kg/m$^2$ (**E,F**) VIL thresholds.

The 20 kg/m$^2$ VIL threshold fields (panels A and B) initially show that the recurrent areas are similar: in particular, the north-central area, clearly influenced by the topography, and the western region, agreeing with previous analysis [11,20]. However, there are two notable differences. First, the southern area had, in 2022, a lower affectation, which is practically null except for some reduced areas. On the contrary, the central coast region had a larger incidence in the spatial distribution. This point could partially explain the

high amount of observations because this is the most densely populated area of Catalonia. In this way, Schuster et al. [40] showed the importance of population density in the hail reports in a territory.

The 40 and 60 kg/m$^2$ thresholds showed a similar behavior. In the case of the 2013–2021 period, the distribution of cases indicates a major incidence in an axis that crosses Catalonia from S to NE, quasi-parallel to the coast. In the case of 2022, the cases are concentrated in central Catalonia and the coastal area. However, the number of cases is not enough to conduct a significant analysis for both thresholds.

### 3.4. Differences in the Monthly Spatial Distribution

As in the case of the hail reports, we performed a monthly analysis to determine if there exists an agreement with the behavior previously detected. Figures 5–7 show the spatial patterns for June, July, and August, respectively. The shape of the fields for the three VIL thresholds is very similar to the yearly map during the period of 2013–2021. On the contrary, only a few cases in the extreme western area were estimated for the 20 and 40 kg/m$^2$ (none for 60 mm) during 2022. This indicates a negative anomaly for 2022 during the first month of the summer, coinciding with the hail reports.

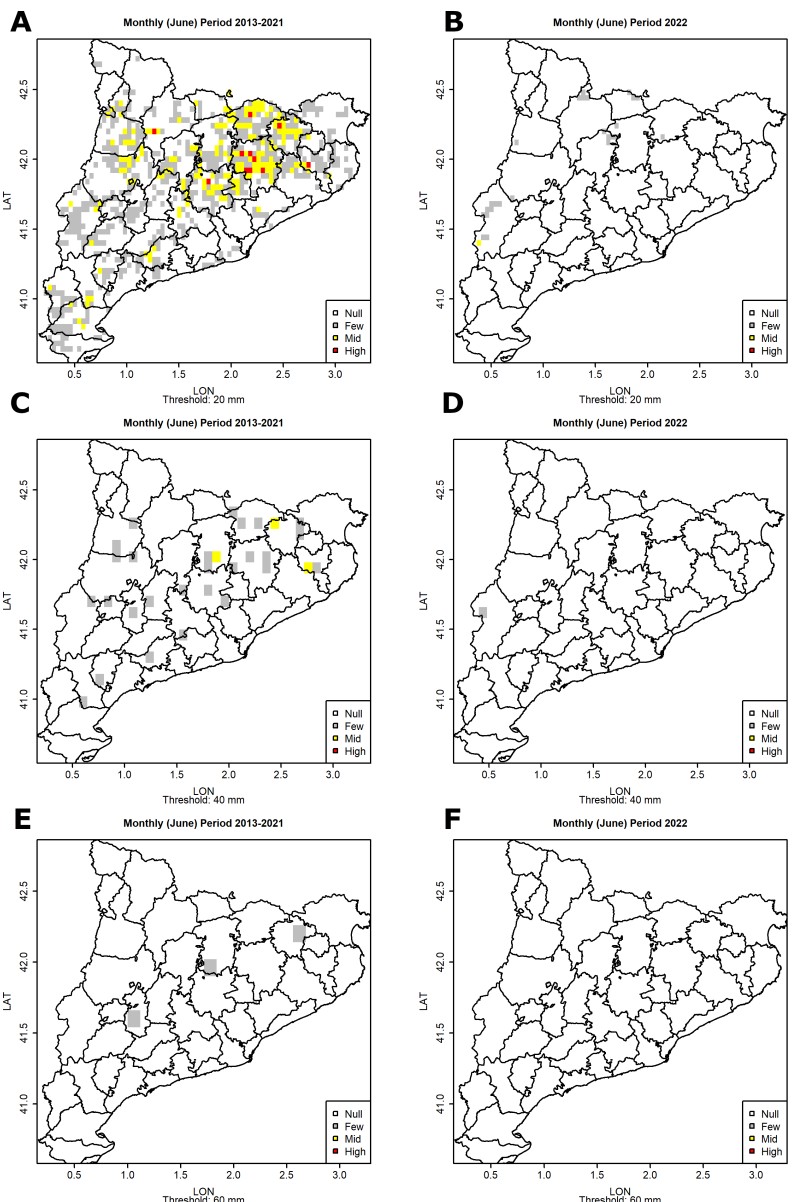

**Figure 5.** (**A**–**F**) Same as Figure 4, but for the June monthly distribution.

July presents the absolute maximum activity in the NE part of the region during 2013–2021, with two secondary maximums in the north-central and the north-western parts. These patterns clearly appear in the 20 kg/m$^2$ threshold, and are less evident in the 40 and 60 kg/m$^2$ thresholds. In the case of 2022, the activity in July was restricted to the central area, with some activity in the north-eastern area. This is a consequence of another month with a low hail occurrence, as the hail reports also show.

Finally, August is the month with the largest agreement between both periods of 2013–2021 and 2022. The pattern is very similar for the three VIL thresholds, with some differences as a consequence of the high variability in the hail events (as they are stochastic random processes). In this case, 2022 presented more activity than usual (2013–2021) in the central coast, and fewer cases in the southern part.

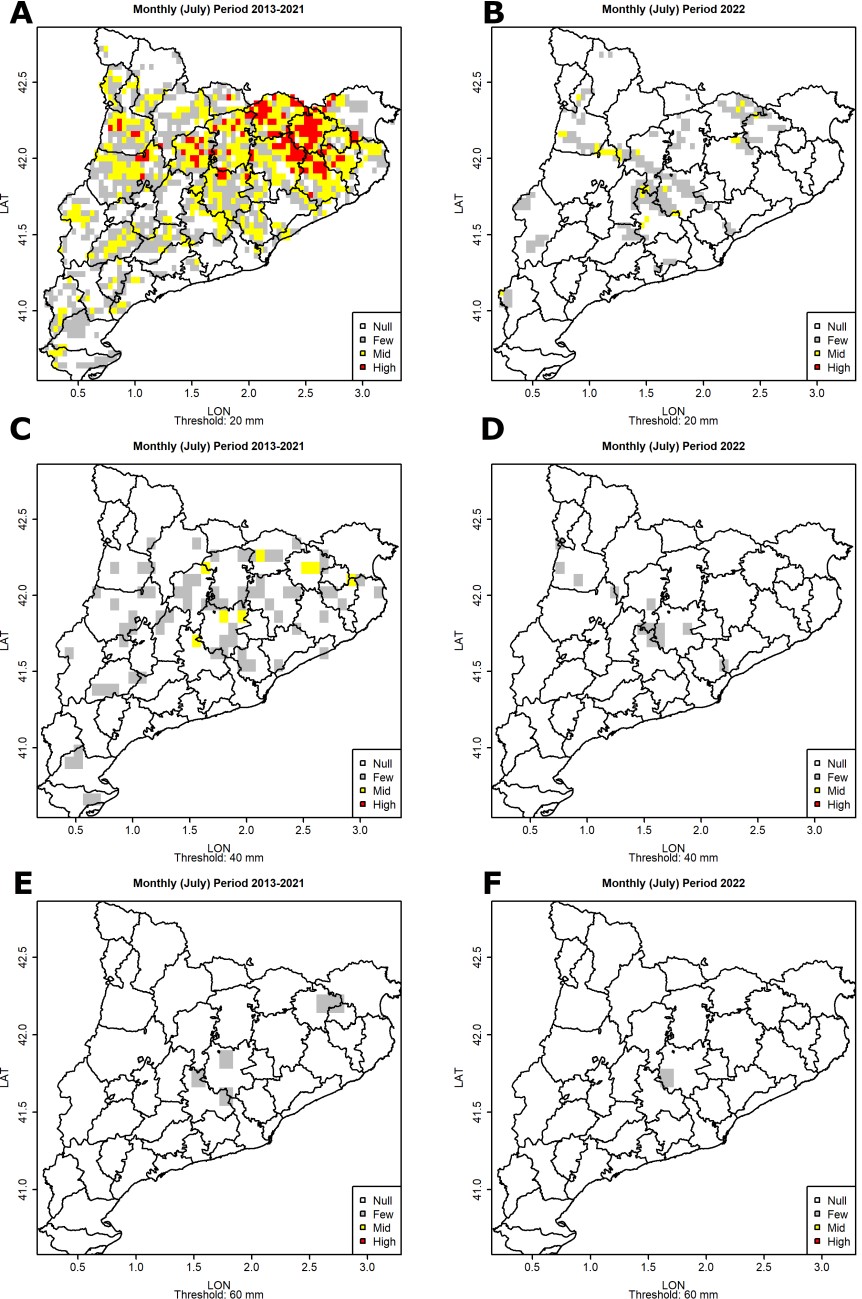

**Figure 6.** (**A–F**) Same as Figure 4, but for the July monthly distribution.

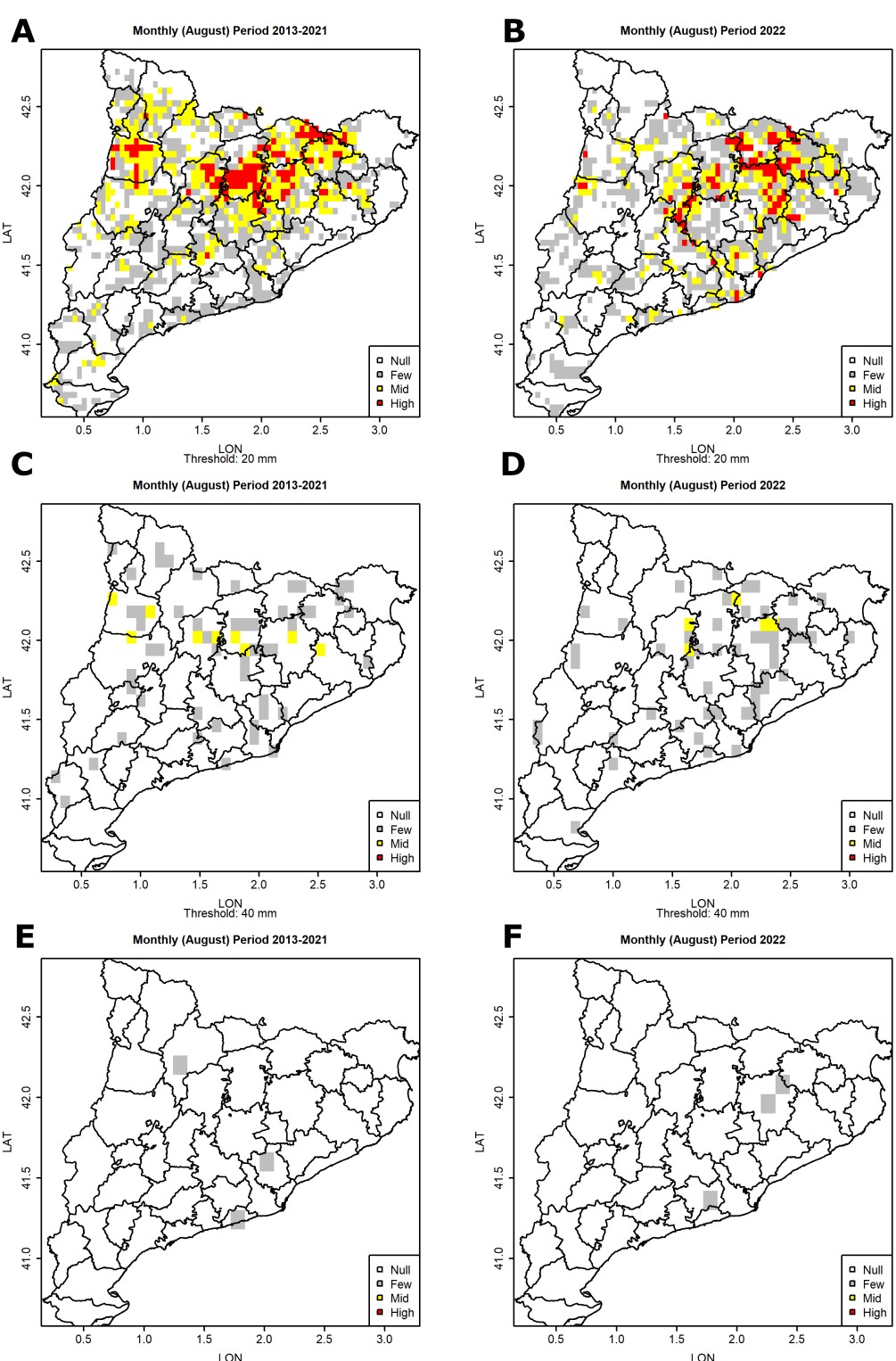

**Figure 7.** (**A**–**F**) Same as Figure 4, but for the August monthly distribution.

To end this section, we analyzed the area of the hail swaths in order to determine if the hit regions were similar in 2022 to those of the period 2013–2021 or, on the contrary, had differences. Figures 8 and 9 show the box plots of the yearly data sets for both periods (grey for 2013–2021 and red for 2022). The area was higher in the case of summer 2022, mainly for the 75th and 90th percentiles, and less significant for the 10th, 25th, and 50th ones.

In the case of the monthly distribution (Figure 9), the box plots of the period 2013–2021 are very similar for all three months, with a P50 of 25 pixels for the three months and a P75

between 50 and 60 pixels. On the contrary, the monthly behavior in 2022 was in accordance with that observed for the hail reports, with low values for June (P50 of 25 and P75 of 50) and higher values for July and August (P50 of 35 and 30, and P75 of 95 and 85, respectively). This indicates that the number of hail swaths in July and August of 2022 was higher than usual, whereas, for June, it was lower.

**N pixels (Yearly)**

**Figure 8.** Estimated area (in number of pixels, where each pixel is equivalent to 1 km × 1 km) for the total set of cases of hail swaths estimated with the 20 kg/m$^2$ VIL threshold, for 2013–2021 (grey box) and 2022 (red box).

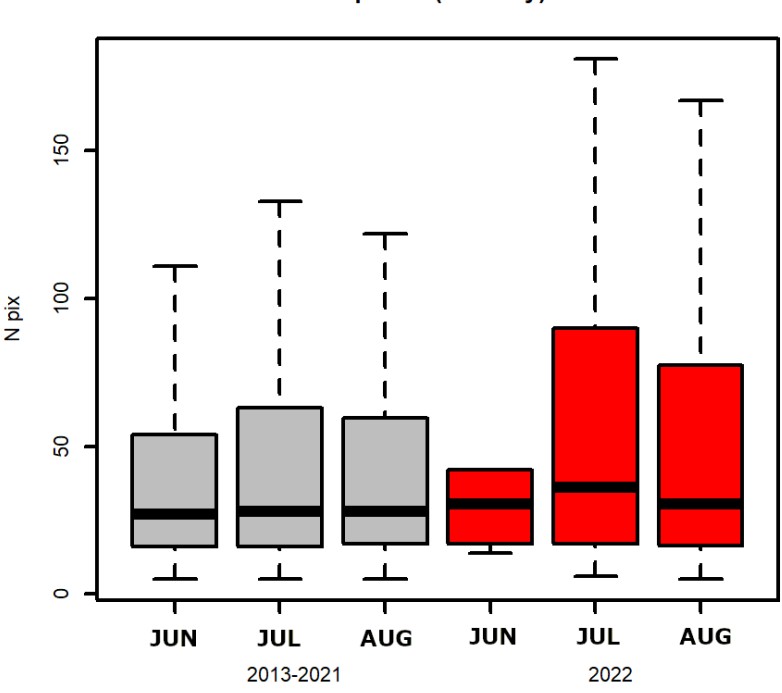

**Figure 9.** Same as Figure 8, but for the monthly cases.

### 3.5. Comparison of Hail Swaths from Radar

The last analysis consisted of a brief comparison of the hail swaths estimated using the radar fields in order to confirm the coherence of both data sets. Figure 10 shows the monthly comparison, for the two periods, considering, in the case of 2013–2021, the averaged values. The differences are notable for the months of June and August, whereas the case of July is similar for both periods. In the case of June, 2022 presented a clear deficit of cases with respect to 2013–2021 (10 cases versus 35). The opposite is the case for August, with 250 hail swaths in the summer of versus 55 for the reference period (2013–2021). This is in accordance with the hail reports shown in the panel A of Figure 3, indicating that the estimations made with the weather radar coincide with the direct observations of hail events. The differences in the values can be associated with different factors, such as the occurrence of events not reported by any spotter.

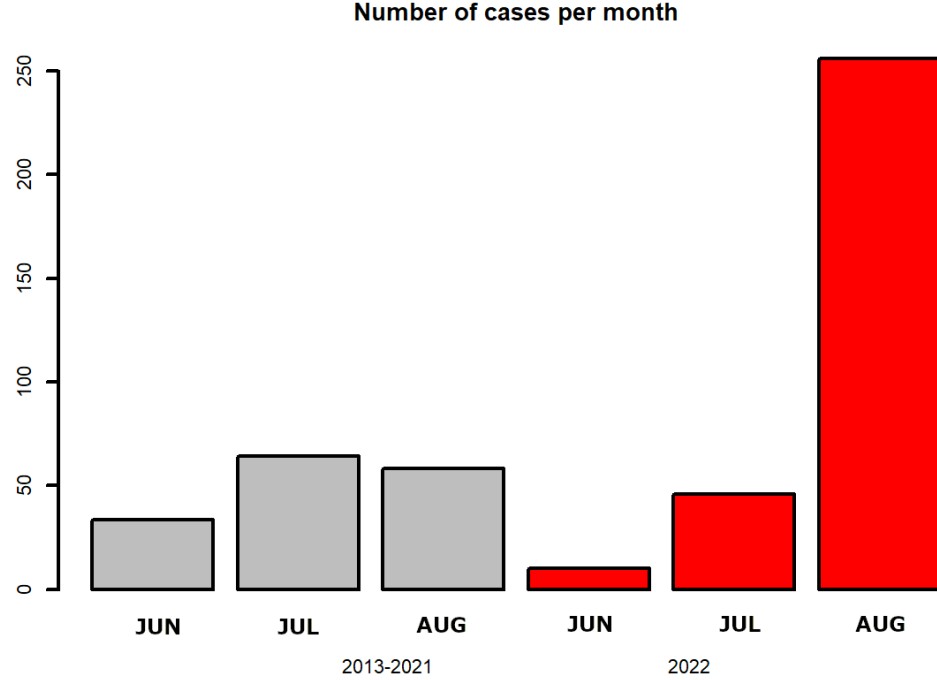

**Figure 10.** Number of cases per month for the two periods (in grey, averaged for 2013–2021, and in red, for 2022) for the hail swaths estimated with the 20 kg/m$^2$ VIL threshold.

### 3.6. Thermodynamic Comparison

In order to determine which meteorological elements contributed to the conditions that led to the occurrence of the elevated number of severe and very large hail, we analyzed five thermodynamic variables. These variables were the convective available potential energy (CAPE), convective inhibition energy (CIN), equilibrium level (EL), lifted index (LI), and maximum updraft strength (WMax). They were selected according to different references that indicate their importance in hail events in Catalonia and surrounding areas [15,16,39]. The parameters were calculated for the Barcelona sounding using the thunder package for R-cran [41]. Table 1 shows the results for the days with hail over 2 cm, comparing the period of 2013–2021 with 2022.

**Table 1.** Thermodynamic variables (from top to bottom: CIN, CAPE, EL, LI, and Wmax) compared for the campaigns of the period of 2013–2021 (left column) and 2022 (right column).

|  | **2013–2021** | **2022** |
|---|---|---|
| CIN (J/kg) | −11.09 | −11.22 |
| CAPE (J/kg) | 2034.9 | 2945.7 |
| EL (m) | 11,787.28 | 12,441.42 |
| LI | −6.01 | −7.90 |
| Wmax (m/s) | 62.49 | 76.76 |

Sanchez et al. [39] observed an increase in the EL (or the height where convection stops) in the Pyrenees area in the last few years. This fact has also been observed in our research, with a gradual increase in the last few years, with the unique exception of 2017, when a secondary peak occurred (not shown). The value for 2022 (12,441 m) is 700 m higher than the average value for 2013–2021, indicating that convection reached higher altitudes and allowed for a major freezing of the updraft. This updraft was also more intense in 2022, with a value of 76.76 m/s, 14 m/s higher, than for the period of 2013–2021. The reason for these differences can be found in the values of CAPE and the LI, which indicate the high amount of available energy and atmospheric instability, respectively, during the summer of 2022. In this sense, CAPE presented a value that was near to 1000 J/kg higher than for 2013–2021, and the LI was 1.9 lower for the same period. All of the parameters were higher or similar to those observed in the analysis of [16] for large hail occurrence. The abnormally long period of warm temperatures caused an extra supply of energy that was triggered in consecutive events at the end of the period. Finally, the inhibition of the energy, represented by the CIN parameter, showed similar values to the other years, indicating that there was practically no constraint to the thunderstorms' development.

## 4. Conclusions

The summer of 2022 will be remembered by the large number of hail cases, but especially by a couple of them that had a high social and media repercussion. The purpose of this research was to confirm whether this year was actually extraordinary or, on the contrary, if it had similar registers to previous years but affected more densely populated areas. It is important to confirm or not the hypotheses of the increase in severe hail events in the context of global warming ([28,39], among others). To achieve this, we used two type of sources. First, hail reports, which have the advantage of being true values (all of the data considered must pass a quality control), but have, as a major issue, the limiting factor that not all of the cases are well-detected (in night episodes or those occurring in remote areas). Second, maximum daily VIL fields, which are indirect estimations but permit good approximations of the hail swaths in most cases, as Rigo and Farnell showed [10].

To apply context, it is important to introduce that the summer of 2022 was one of the warmest ever to occur in Catalonia according to the observations of the SMC. These positive anomalies in temperature were persistent during the three months. In addition, these three months are those with more hail activity in the region according to the SWDB. Finally, the SMC has deployed a VIL database in geotiff format since 2013, which allows for a rapid analysis of the field. Because of this, the period of comparison was 2013–2021. This period is not significant in climatic terms, but it is representative enough of the hail occurrence in the region, as other studies have shown for other areas [35,42].

The hail report analysis revealed that the total number of observations in 2022 was elevated but not extraordinary compared with 2013–2021. In monthly terms, June had much fewer cases, July had an average number, and August presented an exaggerated positive anomaly in all of the studied parameters: the number of observations, the number of hail days (total, severe, and large), and the maximum hail size. This high active period that covered 28 July to 30 August has a clear similarity to that presented by [26] for the

summer of 2021 in Switzerland. This could be a signal that, if the atmospheric conditions are adequate, the persistence of extreme hail events can be common in upcoming years. On the contrary, the number of cases in less favorable conditions could dismiss this, as some analysis previously indicated [28,43].

The analysis of the VIL fields has allowed for the identification of the estimated hail swath areas. Although there could be some differences with the ground truth, the results are acceptable in most cases [10]. In this case, the results are in the same direction as those observed with the hail reports: very few and disperse cases in June, some but very extended events in July, and more than usual events and with large affectation in August. Finally, while the number of hail swaths presented a reduced value for June and a similar-to-average value for July, the number of cases for August was extraordinary, with a five times higher number of cases than the usual. The hail cases have, on average, major severity in August in Catalonia according to the SWDB, but they are less common than in other months from a climatological point of view. Thus, a repetition of the 2022 summer scenario in the future would imply an increase in severe and very large hail events, and a reduction in episodes with small hail, at least in summer. However, we are not able to confirm whether 2022 was a new sign of global warming, or, on the contrary, if it was an anomaly.

To sum up the results of the research, we can conclude that:

- From the previous years, we observed a large number of hail events in June and fewer cases in August. The first cases had a lower size of the hail stones, whereas August events presented more severe hail. In addition, the area of the hail swath was also larger in August events, but the differences were not important on average;
- The summer of 2022 was characterized by extremely hot temperatures and very dry conditions, with an important drought in the entire Catalan territory;
- June and practically all of July of 2022 can be characterized by few cases and hail below 2 cm;
- On the contrary, the last part of summer 2022 (the last four days of July and all of August) presented different behavior, with many very large hail reports, affecting, in most cases, non-usual areas (e.g., coastal zones), with the extent of the hail swaths clearly exceeding the average, even presenting a case of giant hail;
- This unusual situation resulted in many damages to humans and properties, leading us to conclude that it is necessary to improve the warning chain; in particular, the steps directed to the end-users, which is the dissemination of alerts to citizens.

It is worth noting that hail storms have a randomly stochastic component in many senses: the time of day, the month of the year, the maximum and averaged hail size, the abundance, and the length and strength of the hail swath, among others. Thus, statistical analyses are more limited than in other meteorological analyses. This prompts us to ask if the events that occurred in 2022 were an anomaly or if they can be repeated in the upcoming years. In this way, future work will be to conduct the same analysis in the next few years in order to determine if the 2022 hail campaign was an anomaly in time or, on the contrary, if it will be recurrent. In addition, a deeper analysis should be conducted in order to contribute to clarifying potential changes in local micro-physics and/or the synoptic environment. To sum up, this analysis can be indicative of what can be expected in similar regions to Catalonia.

**Author Contributions:** Conceptualization, T.R. and C.F.; methodology, T.R. and C.F.; software, T.R. and C.F.; validation, T.R. and C.F.; formal analysis, T.R. and C.F.; investigation, T.R.; resources, C.F.; data curation, T.R. and C.F.; writing—original draft preparation, T.R.; writing—review and editing, T.R. and C.F.; visualization, T.R. and C.F.; supervision, C.F.; project administration, T.R. All authors have read and agreed to the published version of the manuscript.

**Funding:** This research received no external funding.

**Data Availability Statement:** No data are available.

**Acknowledgments:** The authors want to thank to the Servei Meteorològic de Catalunya for helping in the research.

**Conflicts of Interest:** The authors declare no conflict of interest.

**Abbreviations**

The following abbreviations are used in this manuscript:

| | |
|---|---|
| SWDB | Severe Weather Database of the Servei Meteorològic de Catalunya |
| CDV | Radar of Creu Del Vent peak |
| PBE | Radar of Puig Bernat peak |
| PDA | Radar of Puig d'Arques peak |
| LMI | Radar of La Miranda peak |
| PPI | Plan Position Indicator |
| CAPE | Convective Available Potential Energy |
| CIN | Convective INhibition energy |
| EL | Equilibrium Level |
| LI | Lifted Index |
| WMax | Maximum updraft strength |
| VIL | Vertically Integrated Liquid |
| SMC | Servei Meteorològic de Catalunya (in Catalan, Meteorological Service of Catalonia) |

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
