# Peer review of "A Summary of Hail Events during the Summer of 2022 in Catalonia: A Comparison with the Period of 2013–2021"

_remotesensing, doi:10.3390/rs15041012_

Round 1

Reviewer 1 Report

The article presents an interesting case study of hail in Catalonia in 2022 against the background of the period 2013-2021. It is solidly written, in accordance with the rules of scientific publication. The graphic side and scientific language do not raise objections. Only at the beginning of the results it is worth adding a subchapter on weather conditions generating strong hail, e.g. what atmospheric circulation, air masses, etc. caused heavy hail, especially in August 2022. The article meets the publication requirements.

The article includes a case study analysis of hail in Catalonia in 2022 against the background of the period 2013-2021. It has been proven based on analyzes of ground registers and radar fields as well as statistical analyzes of the extreme significance of hail in 2022 against the background of previous data. The usefulness of data from radar fields for the spatial location of hail was demonstrated.   Although the study is not innovative in terms of methodology, it is characterized by the aspect of novelty in characterizing the spatial variability and dynamics (size) of hail in the research area. And in this area it complements the state of knowledge.   Presentation of the analysis increases the scope of knowledge about the occurrence of dangerous atmospheric phenomena (hail) in the research area. Indicates new empirical values for the frequency of occurrence of this weather phenomenon and the size of hail. It therefore presents new threshold values for hail, which have been confirmed by data from radar fields and ground registers and by means of statistical analyses. ad 4) What specific improvements should the authors consider regarding the
I have no critical remarks as to the methodology of the proceedings. I would only add the meteorological background of the occurrence of extreme hail in 2022, i.e. what meteorological situation caused these events (atmospheric circulation, air masses, atmospheric fronts, etc.). Such a meteorological background would significantly supplement the conditions for the occurrence of hail.
  The conclusions are consistent, specific and well researched. They clearly answer the presented research problem.
  References are appropriate and sufficient.   The engravings were made very carefully. The research results are very well presented. The number and form of the figures is appropriate.

Author Response

Dear Reviewer,

Please, find attached our answers to your report.

Best regards

Reviewer 2 Report

Dear authors,

The comments and suggestions are attached,

Best redards.

Author Response

(The authors gave the same response as above.)

Reviewer 3 Report

Review of “A Summary of Hail Events during the Summer of 2022 in Catalonia: A Comparison with the period 2013-2021” submitted to Remote Sensing.

GENERAL COMMENTS:

This is an interesting study comparing an unusual period of hailstorm activity in Catalonia (three summer months in 2022) to summertime hailstorm activity in prior years. The analysis methodology and results are rather basic (i.e., not very elaborate), but scientifically sound. However, given the primary focus involves comparing two data sets, some quantitative statistical analysis would be useful. The only major problem I have with the article relates to the need for extensive editing/improvement of the English text. Although I didn't have great difficulty understanding what the authors were trying to say, in my opinion, the grammatical quality of the current article is very poor.

SCIENTIFIC COMMENTS:

1. Line 41: I assume that “ground registers” refers to hail observations/reports. If so, then for clarity, it would be good to mention that here (I do see mention of this later in section 2.2.1), or perhaps simply replaceground registers” with “hail reports”.

2. Line 51: Please clarify what you mean by “modeling the atmospheric conditions”. I assume you're not talking about numerical weather prediction.

3. Lines 92-95: Beyond simply comparing the data sets via box plots, quantitative statistical tests should be used (e.g., Student's t-test or Wilcoxon test) and the results presented.

4. Lines 99-100: Why are the units of VIL shown as “mm”? Every article or paper I've read involving VIL has its units as “kg m-2”, including references [32] and [33].

5. Line 187: By “similar behavior” do you mean not much difference (for the two periods of 40 and 60), or similar to the patterns in A and B?

6. Line 199: Add “for 2022” after “anomaly”.

7. Lines 218-219: In Fig. 8, the only meaningful difference I see is with the upper quartile (P75) and the upper adjacent value (line above P75). The others are nearly identical.

8. Lines 229-239: This section is confusing, as it's not clear what is being shown in Fig. 10 and discussed in the text. The caption of Fig. 10 says hail-swaths estimated from VIL, but nothing is mentioned in the text (lines 231-232) about hail-swaths or VIL. And line 235 says “250 registers”, which suggests hail reports. Lastly, what do you mean by “in front of” on lines 235 and 236?

9. Line 292: “few cases in August” is not technically correct based on Fig 3C, where there is a lot of overlap for the boxes, especially the large hail days. Fix this by changing “few” to “fewer”.

STYLISTIC COMMENTS:

1. Line 1: Replace “usual” with “common”.

2. Line 47: Replace “Materials” with “Data”.

3. Lines 92, 141, 164, 166, 267, 284, 300 and Fig. 2 caption: I suggest replacing “large hail” with “very large hail”, as many other studies equate large hail with severe hail.

4. Line 125: Replace “to” with “from”.

5. Fig. 3 : Either the x-axis labels should be changed to better clarify what is being shown (e.g., in A, replace “06” with “Jun”) or the caption be modified to directly link the labels to their actual meaning. Also, in A and B, you might want to use a different color box than yellow, which is difficult to see on a white background. And, you need to mention in the sentence “The light grey boxes correspond ...” that this refers to A and B.

6. Line 154: Replace “in” with “at”.

7. Line 178: Replace “coincident” with “similar”.

8. Line 222: Replace “both” with “all three”.

9. Lines 225-227: Remove multiple use of “much”, as the differences in P50 are not very large.

10. Figs. 9 and 10 : As with Fig. 3, either the x-axis labels should be changed to better clarify what is being shown (e.g., replace “06” with “Jun”) or the caption be modified to directly link the labels to their actual meaning.

11. Fig. 10 caption: Replace “Distribution” with “Number” to match the title above the figure.

12. Line 255: Replace “registers” with “observations”.

Author Response

(The authors gave the same response as above.)

Reviewer 4 Report

I reviewed the paper titled “Dynamic Downscaling GCM Result to Analyze Long-term Climate Change in Hunan Province, China”. Below are my comments on the paper.

Line 6, what is VIL?

Line 6, it says “ these regions”, what are these regions? There is no previous knowledge about any regions.

The abstract does not provide a specific results obtained from the study nor the conclusion and recommendations.

The introduction is not concrete. In which, what is the difference between this study and others.

The paper novelty and objectives are not explicitly presented.

Line 89 syas “ the following steps” but the next sentences nor paragraph has not mentioned any steps. Please clarify.

Section 2 “data and methods” is vey short and little confusing. I have not fully understood it. a general flowchart can ease the redear to understand the methodology.

Line 136, how did the authors noticed if there is no official confirmation?

The results indicated some differences between the summer 2022 comparing with previous years 2013-2012, are these differences statistically significant?

In conclusions are little vague. What are the future application of this study?

Also, I am wondering about the paper novelty where the papers seems using an existing approach without any further modification/development.

Author Response

(The authors gave the same response as above.)

Round 2

Reviewer 2 Report

Dear Authors,

My considerations are attached.

Best regards,

Author Response

Dear Reviewer,

Please, find attached the answers to your review.

Best regards

The Authors

Reviewer 3 Report

Review of revised article: “A Summary of Hail Events during the Summer of 2022 in Catalonia: A Comparison with the period 2013-2021” submitted to Remote Sensing.

GENERAL COMMENTS:

I appreciate the authors incorporating my suggested modifications to the article. As such, it is notably improved from the original version. However, I still remain concerned about the quality of the English text, and feel that further improvement of the article is possible in that regard. At this point, I leave it to the editor to decide on the amount of editing of the English language and style that is required prior to final acceptance and publication.

SCIENTIFIC COMMENTS:

1. Lines 91-99: I would suggest that within or at the end of this new paragraph (that explains in greater detail VIL), to include one or two references to earlier articles that go into more extensive discussion on the calculation of VIL, such as:

Amburn, S. A., and Wolf P. L. , 1997: VIL density as a hail indicator. Wea. Forecasting, 12, 473–478.

Greene, D. R., and R. A. Clark, 1972: Vertically integrated liquid water - A new analysis tool. Mon. Wea. Rev.,100, 548–552.

2. Lines 111-115: I appreciate the inclusion of quantitative statistical results. However, these results should be presented somewhere in Section 3, not in Section 2.3. If desired, you could mention here that quantitative statistical testing will also be performed (involving the t-test).

3. Line 256: There remains an inconsistency between saying 250 “hail reports” while Fig. 10 shows hail-swaths estimated using VIL.

STYLISTIC COMMENTS:

1. Line 102: Replace “hail” with “Hail”.

2. Line 146: Replace “for” with “from”.

3. Line 243: Remove “both”.

4. Lines 256 and 257: Replace “in front of the” with “versus”.

5. Line 280: Add “higher” before “than”.

6. Lines 307, 315 and 318: Replace “registers” with “observations”.

7. Line 352: Replace “registers” with “reports”.

Author Response

(The authors gave the same response as above.)
